# Thermal comfort perception among park users in Prague, Central Europe on hot summer days—A comparison of thermal indices

Vlaďka Kirschner[1]*, Aleš Urban[2,3], Lucie Chlapcová[2], Veronika Řezáčová[4]

1 Department of Landscape and Urban Planning, Faculty of Environmental Sciences, Czech University of Life Sciences Prague, Prague, Suchdol, Czech Republic, 2 Department of Water Resources and Environmental Modeling, Faculty of Environmental Sciences, Czech University of Life Sciences Prague, Prague, Suchdol, Czech Republic, 3 Institute of Atmospheric Physics of the Czech Academy of Sciences, Prague, Záběhlice, Czech Republic, 4 Crop Research Institute, Prague, Ruzyně, Czech Republic

* kirschner@fzp.czu.cz

**Data Availability Statement:** All relevant data are within the paper and its Supporting information files.

**Funding:** VŘ was supported by the Ministry of Agriculture of the Czech Republic, institutional

## Abstract

The assessment of human perception of the thermal environment is becoming highly relevant in the context of global climate change and its impact on public health. In this study, we aimed to evaluate the suitability of the use of four frequently used thermal comfort indices (thermal indices)–Wet Bulb Global Temperature (WGBT), Heat Index (HI), Physiologically Equivalent Temperature (PET), and Universal Thermal Climate Index (UTCI)–to assess human thermal comfort perception in three large urban parks in Central Europe, using Prague, the capital of the Czech Republic, as a case study. We investigated the relationship between the four indices and the thermal perception of park visitors, while taking into account the effect of the sex, age, and activity of the respondents and the week-time and daytime of their visit (assessed parameters). Park visitors were interviewed during the summertime, while collecting meteorological data. The correlations were performed to explore the relationship between the thermal perception and the individual thermal indices, multivariate statistical methods were used to explain how well the variation in thermal perception can be explained by the assessed parameters. We found a significant association between all the indices and thermal perception; however, the relationship was the strongest with HI. While thermal perception was independent of sex and week-time, we found a significant effect of age, physical activity, and daytime of the visit. Nevertheless, the effects can largely be explained by thermal conditions. Based on the results, we conclude that all the investigated indices are suitable for use in studies of thermal comfort in parks in Central Europe in summertime, while HI seems the most suitable for architects and planners.

## Introduction

Climate change projections indicate a likely increase in temperatures and in the intensity of urban heat islands worldwide [1], thus amplifying the risk to public health [2]. Therefore,

support MZE-RO0423. AU was supported by the Czech Academy of Sciences programme "Strategie AV21 – Dynamická planeta Země".

**Competing interests:** The authors have declared that no competing interests exist.

heightened attention has been dedicated to the adaptation of urban outdoor spaces to the changing conditions and to human perception of them [3–5]. Nature-based solutions, such as green and blue infrastructure, are widely accepted measures which substantially influence local microclimate and thermal comfort [6]. Climate-sensitive urban planning aims to integrate climate information with urban planning in order to improve outdoor thermal conditions in cities and thus create a thermally comfortable living environment [7].

Thermal comfort indices (thermal indices) have become a popular tool for assessing the outdoor environment. The indices aim to illustrate how weather and climate affect people in urban areas [8]. More than 160 outdoor thermal indices were created across the world [9]. There are indices that are relatively simple, while others are more comprehensive. The relatively simple indices are based on atmospheric measures related to the environment's impact on heat gain, such as air temperature, mean radiant temperature, relative humidity, and wind speed [10]. They can be represented by Wet Bulb Global Temperature (WGBT) and Heat Index (HI). The HI is solely based on air temperature and relative humidity. WBGT combines measurements from three thermometers (black globe, wet bulb, and dry bulb) to account for air temperature, humidity, and the impact of solar radiation on heat gain [11]. It also indirectly accounts for the effects of wind. Both WGBT and HI can be calculated directly by portable thermometers, such as Kestrel's Heat Stress Tracker (https://kestrelinstruments.com), which is often used because it is relatively inexpensive and simple [12].

More comprehensive indices, such as Universal Thermal Climate Index (UTCI) and Physiologically Equivalent Temperature (PET), aim to describe the physiological responses to multi-dimensional outdoor thermal environments. They are based on human energy balance models [13, 14] rising from the assumption that the perception of thermal comfort is affected by the entire set of person-related aspects, including the physical, physiological, psychological, and socio-behavioural aspects underpinning thermal comfort [9, 15, 16]. For these reasons, people in different geographical regions adapt differently to the thermal environment [6, 17]. For instance, people from Freiburg feel comfortable at 13˚C, while people in Athens feel comfortable at 23˚C [7]. For calculations, the UTCI and PET use the above-mentioned atmospheric measures (air temperature, humidity, wind speed, and solar radiation) and incorporate them into a multi-node model of human thermoregulation (for UTCI: [18] with considerations for metabolic workloads and clothing type (for UTCI [19]). In order to calculate the comprehensive indices, several microclimate models have been developed [20]. The RayMan model (developed by [21]) and the advanced SkyHelios model (developed by [22]) have often been used to calculate both UTCI and PET. UTCI can also be calculated directly on utci.org.

HI, WGBT, UTCI, and PET are among the most popular indices [6, 11, 23–25], while the use of UTCI, being relatively new (proposed by [19]), has been increasingly employed over the past decade, especially in European studies [11]. It has often been used in health-related studies [2, 26]. Recently, UTCI use increased due to the development of the UTCI global dataset in 2021 [27]. PET, developed in 1999 by Höppe [28], has been widely used for thermal comfort assessment in various outdoor urban settings (e.g. [3] in Shanghai; [4] in Chile; [29] in Rome). The oldest index is the WGBT, developed in 1954 [30]. It was proposed to improve the prediction accuracy of thermal comfort amongst American military members, lately, in 1996, and it was applied to prevent heat injury in the U.S. [6]. The HI was developed in 1984 [31], and it became widely used in weather services and weather forecasts in the U.S. [11].

The urban spaces, designed by architects, landscape architects, and planned by urban planners, are generally considered to be well-designed and well-planned when used by people [32]. Therefore, well-connected and well-equipped urban spaces [33, 34] have been considered to be of a high quality. In the time of ongoing climate change [1], healthy and liveable urban

conditions defined by the thermal comfort of inhabitants have become an essential issue for urban design and planning [35, 36]. Urban planners and designers have been encouraged to use thermal indices to make decisions regarding the proper design of urban configuration [17, 37, 38]. However, architects and planners often neglect thermal comfort because of their limited knowledge about the indices' suitability and ability to calculate them [24, 39]. Therefore, there is a need to define an index that would help architects and planners to create a thermally comfortable living environment based on people's needs and perceptions. Such a need has also been noticed by other researchers, who reacted by proposing new indices (e.g., NETCID [24]; TSI [40]). As there have been more than 160 indices in the course already [9], we find it more appropriate to identify the most suitable one from the existing indices. We believe that such an index should be associated with people's perceptions in order to create urban spaces to be used by people.

## Parameters affecting thermal perception

Microclimate parameters affect objective and subjective comfort only by 50% [41]. There have been several studies pointing out potentially influential parameters. Some studies [29, 42] mention the intensity of physical activity as influential for thermal perception. With increasing physical activity, the metabolic rate increases [29] as well as the perception of thermal discomfort [43]. Furthermore, the intensity of outdoor physical activity (such as sitting, recreational walking and running) can be influenced by different landscape patterns [35, 43, 44]; and the activity may differ depending on the daytime [33, 42] or week-time [33]. For instance, children do more dynamic activities than the elderly, and do them more often in the afternoons [33]. Adjusting the activity rate and the time can help in coping with thermal changes [43].

Additionally, people's personal characteristics, such as sex and age, are mentioned in many studies [23, 45, 46], although with inconclusive results. Indeed, while some studies found age to be a significant parameter [42, 47, 48], other studies found that age had a minimal influence on thermal perception [49]. For instance, in Wuhan, China, the elderly were less tolerant of cold compared with young adults and children [42]. Some studies found women to be more sensitive to thermal perception than men [36, 49], some studies found the opposite [50, 51], and some did not find a conclusive relationship between sex and thermal perception [19, 47, 52]. Overall, discrepancies suggest that some other influential factors could be hidden behind the parameters of age and gender. A study about tourists in Warsaw, Poland suggests that people's place of origin could be one of the influential factors, pointing out seven days as a minimal period of acclimatisation [49].

### Objectives

In this study, we aimed to evaluate the suitability of four frequently used thermal indices–HI, WGBT, UTCI, and PET–to assess thermal perception in urban parks in Central Europe with a temperate climate, using Prague, the capital of the Czech Republic, as a case study area. Specifically, we looked for an association between the thermal indices and thermal perception of park visitors, and investigated whether or not thermal perception varies with regards to the sex and age of visitors (sex, age), the visitors' activity (activity), and between weekdays and weekends (week-time), morning, afternoon, and evening (daytime). We hypothesised that (i) all the indices would be associated with subjective thermal comfort, while the more complex indices which reflect a (universal) man (UTCI and PET) within their calculation would be more closely associated with thermal perception than the more simple indices (WBGT and HI); (ii) thermal perception would be affected by sex, age, or the week-time or the daytime.

## Data and methods

The data were collected in a field survey in three parks, observing and interviewing the visitors while collecting meteorological data. UTCI and PET indices were calculated, and all the data were linked for statistical analysis. The methods are described in more detail below.

### Study area

Prague is by far the largest city in the Czech Republic. Located in one of the warmest parts of the country (50˚5′N, 14˚25′E), a warm temperate climate with cold winters and warm summers (Cfb) is typical in this region according to the updated Köppen-Geiger classification [53]. Enhanced by ragged topography, Prague's urban heat island is well developed [54, 55]. While on winter days the urban heat island moderates the effect of cold stress, it may exacerbate the health impacts of heat on hot summer days compared to the surrounding rural regions [56].

Three parks in Prague (Fig 1) were selected for the site survey: Royal Preserve (*Královská obora*, 50˚06'26.6"N 14˚25'19.9"E), Central Park (*Centrální park*, 50˚02'54.5"N 14˚20'17.5"E), and Hvězda Preserve (*Obora Hvězda*, 50˚04'59"N 14˚19'38.9"E). All of these are large parks (approximately 95ha, 40ha, and 80ha, respectively) and are relatively flat, with landscape patterns consisting of a combination of lawns and tree-covered areas; Royal Preserve and Central Park include lakes. Royal Preserve is surrounded by compact block houses, Central Park by blocks of flats, and Hvězda Preserve mainly by family houses (Fig 1). Thus, the surroundings of the three parks represent the three main types of housing forms in Prague.

### Field survey

The field survey was conducted in June and July of both 2022 and 2023, focusing on the warm summer months. The week-time and daytime periods of the survey covered the whole week and the day from 9:30 am to 6:30 pm, aiming to obtain weekly and daily patterns of use. The week-time period was divided into weekdays (from Monday to Friday) and weekends (Saturday and Sunday). The daytime periods were separated into three different categories: morning (9:30 am–12:00 pm), afternoon (12:05 pm–3:55 pm), and evening (4:00 pm–6:30 pm).

Interviews with randomly selected visitors were conducted to obtain information about their thermal perception. No interviews permit was not required by the Disciplinary Committee of Faculty of Environmental Sciences of the Czech University of Life Sciences Prague because it was fully in accordance with the Disciplinary Code of the Faculty and the University. All data was collected anonymously. The park visitors were informed about the nature of the study before the actual interview was conducted, and they were asked to provide verbal consent to use their answers in this study. Only the answers of the visitors who provided verbal consent were used in this study.

Thermal perception is represented by answers to the closed-ended question, "How are you feeling at this moment in this weather?" Respondents could choose from a set of options based on a scale of 1 to 5, with 1 corresponding to the most comfortable and 5 to the least comfortable. Other studies have used a 5-point scale [7], a 7-point scale [6, 46], or a 9-point scale [57] to gauge thermal perception. We chose 5-point scaling in accordance with the school grading system of the Czech Republic, therefore making it easily comprehensible for the Czech respondents. If a group of respondents answered, we recorded only the first two answers, and if the answers were the same, we recorded only one, so that the answers could not be influenced by each other [58]. Considering possible discrepancies arising from individuals being adapted to different climates [29], foreigners (speaking a foreign language or with a foreign accent) answered one more question about the length of their stay in the Czech Republic. An

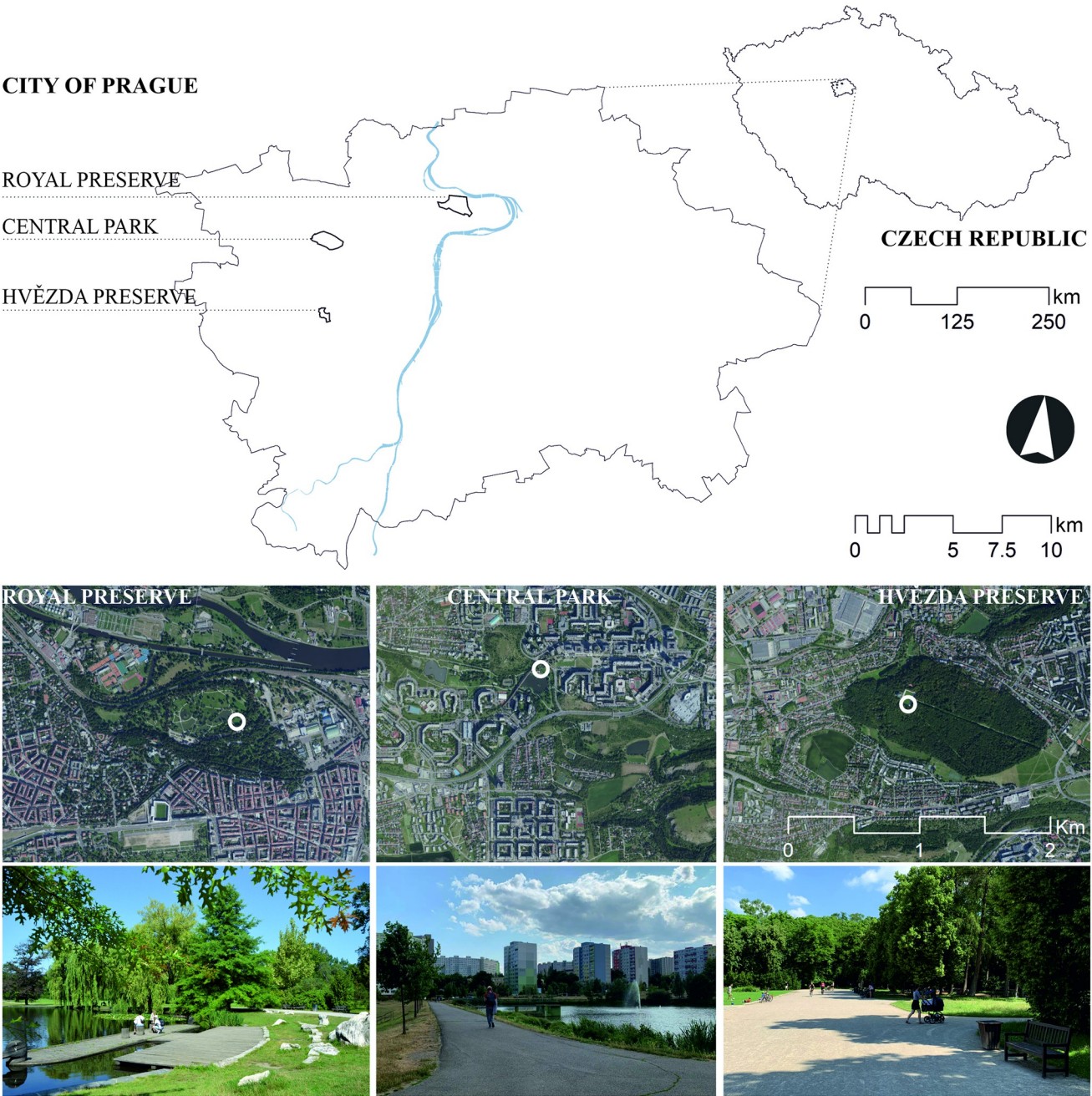

**Fig 1. Study areas: Royal Preserve, Central Park, and Hvězda Preserve in Prague.** The orthophoto maps show the landscape patter of three parks and the surrounding urban forms. The white circles at orthophoto maps define the location of the site survey. The photos were taken during the site survey in July 2023. The maps with the satellite images are openly available on Czech Office for Surveying, Mapping and Cadastre [https://geoportal.cuzk.cz].

additional (optional) explanation of the reason for the thermal perception was recorded in order to better understand the possible sources of thermal comfort or discomfort.

The description of each visitor interviewed was observed by the interviewer. The sex and age were estimated. The sex parameter was divided into "woman" and "man". The age was rounded to 5, and only adults were interviewed. The age parameter was divided into "young"

**Table 1. Assessed parameters observed in three parks during the site survey—In number of respondents.**

| Parameters / park name | Royal Pres. | Central Park | Hvězda Pres. |
|---|---|---|---|
| WEEK-TIME: weekdays / weekends | 115 / 72 | 102 / 81 | 82 / 148 |
| DAYTIME: morning / afternoon / evening | 58 / 68 / 61 | 82 / 67 / 34 | 58/ 57 / 115 |
| SEX: woman / man | 117 / 70 | 109 / 74 | 113 / 117 |
| AGE: young / middle-aged / elderly | 131 / 39 / 17 | 85 / 54 / 44 | 132 / 70 / 28 |
| ACTIVITY: low-active / high-active | 155 / 32 | 173 / 10 | 180 / 50 |

During the site survey, meteorological data were measured by the Kestrel 5400 Heat Stress Tracker (https://kestrelinstruments.com), which was placed on a tripod 150 cm above ground level. The frequency of the measurement was set at 5-minute intervals. Some measured meteorological data (air temperature—TEMP, humidity, wind speed) are displayed in Supplement (S1 Fig) to describe the meteorological conditions in three parks during various days of the survey.

(up to 30 years old), "middle-aged" (approximately 35 to 60 years old), and "elderly" (65 years and above) [59]. The respondent's activity just before being approached was recorded, describing the level of activity as sitting, walking, cycling, or running. Then, because only a minimum amount of persons interviewed were sitting or cycling, we split the respondents into two categories: low-intensive physical activities, "low-active" (including sitting and walking activities) and high-intensive physical activities, "high-active" (including cycling and running).

Overall, 615 interviews were conducted in all three parks, while 15 respondents were excluded from the analysis because they were deemed likely to be influenced by the fact that they were foreigners living in the Czech Republic for less than a year [50] or they had just arrived from abroad (and specifically mentioned this fact in relation to their perception) [49]. Consequently, 600 interviews were considered for the final analyses; 187 were conducted in Royal Preserve, 183 in Central Park, and 230 in Hvězda Preserve. The parameters of the respondents and their visits are presented in Table 1. The number of respondents was relatively evenly distributed according to the week-time and daytime and to the sex of the respondents. More than half of the respondents were young people (348), while the lowest number of respondents were of an elderly age (89). The predominant activity was walking.

## Indices calculation

Four thermal indices were calculated from the measurements. WBGT [30] and HI [31] are two direct indices calculated directly by the Kestrel 5400 device. Two thermal comfort indices, PET [28] and UTCI [18], were calculated from the data recorded using the RayMan Pro software developed by Matzarakis et al. [21]. Dry bulb temperature, relative humidity, globe temperature, and wind speed in height of 1.5 m, measured by the Kestrel device, were used for UTCI and PET calculation to avoid some inaccuracies resulting from required approximation of the RayMan input data [26]. Globe temperature data were used for the mean radian temperature calculation, according to the formula in Kántor and Unger [60]. In addition to meteorological variables, the physiological parameters of the reference person are taken into account in the calculation based on reference values of clothing level, metabolic rate, age, and weight based on the Munich Energy-balance Model for Individuals (MEMI; meaning: height 1.75 m, weight 75 kg, age 35 years, sex–male, clothing 0.9, activity 80 W, position–standing) for PET [61] and Fiala model (height 1.75 m, weight 75 kg, age 35 years, sex–male, adaptive clothing model, activity 135 W–walking) for UTCI [18].

Air temperatures during the survey covered a wide range of temperatures (Fig 2), ranging from 18.2˚C to 42.7˚C. HI ranged from 16.8˚C to 45.5˚C, which covers three levels of the U.S.

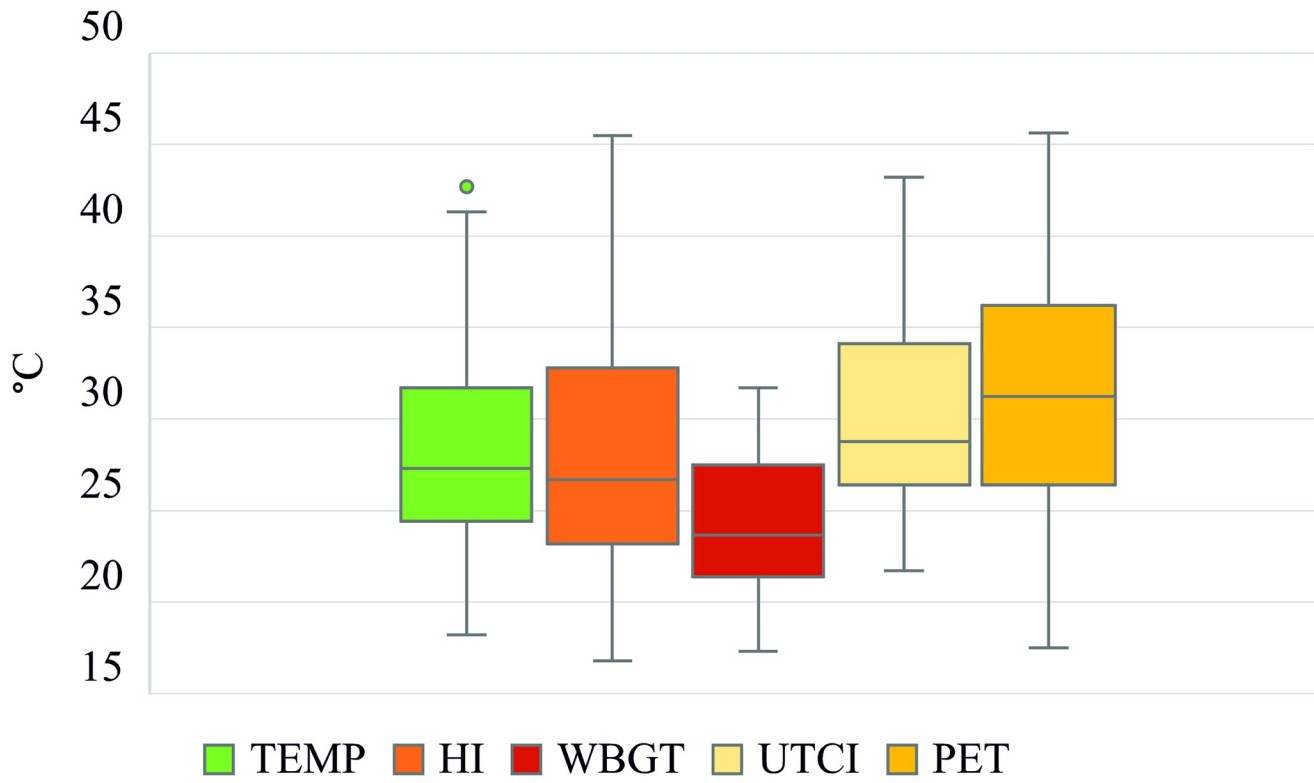

**Fig 2. Distribution of air temperature (TEMP) and the four thermal indices during field surveys in all three parks.**

National Weather Service's heat warning (caution, extreme caution, and danger: https://www.weather.gov/bgm/heat); it does not cover the extreme danger level (from 54˚C), which is a condition that does not apply to Central Europe. WBGT ranges from 17.3˚C to 31.5˚C, corresponding to all categories (from normal to extreme conditions), as defined by Regional heat safety activity guidelines based on the Georgia High School Association policy [62]. UTCI ranges from 20.9˚C to 43.2˚C, covering four caution categories (no thermal stress, moderate heat stress, strong heat stress, very strong heat stress) out of five warning categories in warm conditions (https://climate-adapt.eea.europa.eu/en/metadata/indicators/thermal-comfort-indices-universal-thermal-climate-index-1979-2019); it does not cover extreme heat stress (above 46˚C). PET ranges from 17.5˚C to 45.6˚C, covering six out of nine overall categories (slightly cool, comfortable, slightly warm, warm, hot, and very hot in [28]).

## Data analyses

The basic statistical values comprised averages, and Pearson's correlation coefficients were calculated in Microsoft Excel (Microsoft Corporation, Redmond, WA, USA) after checking for data normality. The correlations were performed to explore the relationship between the thermal perception and the individual thermal indices.

Furthermore, we tested how well the variation in thermal perception can be explained by the assessed parameters—weekday, weekend (week-time), morning, afternoon, evening (day-time), woman, man (sex), young, middle-aged, elderly (age), and low-active, high-active (activity). To assess and address any potential multicollinearity among the parameters included in the analysis, we first conducted detrended correspondence analysis (DCA). Since the assessed

parameters based on DCA (data not shown) did not show any strong collinearity, we proceeded to test the impact of thermal perception on the assessed parameters. This was achieved using canonical correspondence analysis (CCA) supported by Monte Carlo permutation test with significance estimates adjusted using the false discovery rate approach [63] carried out in Canoco 5, which enabled the testing of the effect of all the factors at once. The pack of the analysed data contained 600 respondents (samples) and all the assessed parameters (species; see above), which were as a qualitative (categorical) data coded as dummy (0/1) variables, and one explanatory variable (thermal perception). When using thermal perception as an explanatory, we were then able to construct a T-value biplot showing which of the tested factors were significantly ($p < 0.05$) associated with this variable.

To understand the reason for the significant association between the daytime and the thermal indices, an analysis of variance with $p < 0.05$ as the significance cut-off level was performed to answer the question of whether there was a significant variability in the values of the thermal index that correlated best with thermal perception between morning, afternoon and evening (daytime). The analysis of variance was calculated in R 4.3.1 statistical environment (R Core Team, 2013, http://www.R-project.org/).

## Results

### Associations of the four thermal indices with thermal perception

We correlated the four thermal indices with thermal perception to explore the relationship between them. We found that thermal perception was significantly ($P < 0.05$) positively correlated with all four thermal indices (Table 2), while the correlation was strongest for HI, followed by UTCI, PET, and WBGT. Additionally, all the indices were highly positively correlated with each other (Table 2, not highlighted values).

### Association of sex, age, week-time, daytime, and activity on thermal perception

The association of thermal perception and the assessed parameters is displayed in Fig 3. Thermal perception significantly (Monte Carlo permutation test, pseudo-F = 5.1, p = 0.002) influenced the variability in data of the assessed parameters. In other words, a significant relationship exists between assessed parameters and thermal perception. However, only four parameters reasonably (significantly at $p < 0.05$) explained the variability in thermal perception: middle-aged and low-active respondent, and morning and evening visits (daytime). The significant relationship between the four parameters and thermal perception is indicated in Fig 3. Thermal perception was positively associated with evening visits and low-active respondents, while it was negatively associated with morning visits and the middle-aged group of respondents.

To explain the variability in thermal indices' values during daytime, the thermal indices' (temperature) values in the morning, afternoon, and evening were considered as a potential

**Table 2. Correlation coefficients of the thermal indices and thermal perception (in bold), and the indices with each other.**

|  | HI | WBGT | UTCI | PET | Thermal perception |
|---|---|---|---|---|---|
| HI | 1 |  |  |  |  |
| WBGT | 0.8771 | 1 |  |  |  |
| UTCI | 0.9524 | 0.9444 | 1 |  |  |
| PET | 0.9292 | 0.9304 | 0.9766 | 1 |  |
| Thermal perception | **0.4799** | **0.4276** | **0.4734** | **0.4713** | 1 |

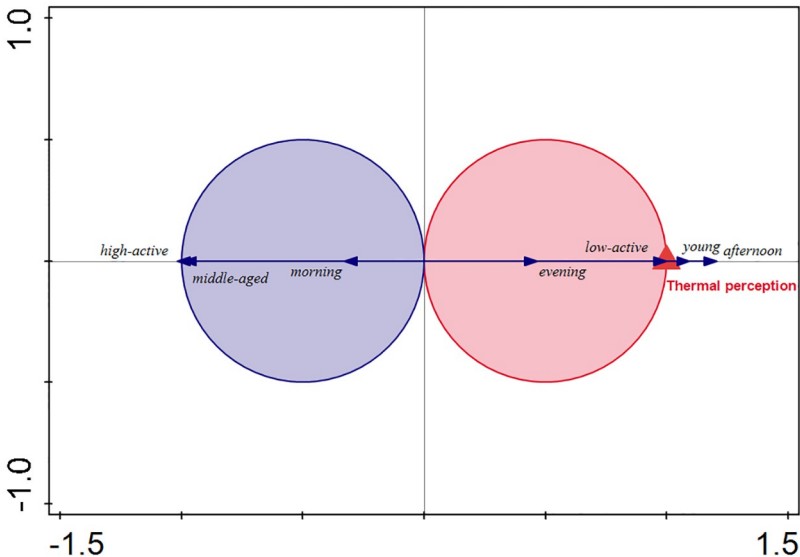

**Fig 3. T-value biplot for relationships between thermal perception and the assessed parameters.** The parameters of the visit–weekday, weekend (time of the week-time), morning, afternoon, evening (daytime), and the visitor–woman, man (sex), young, middle-aged, elderly (age), and low-active, high-active (activity) were assessed. The arrows falling entirely within a Van Dobben circle indicate a significant (p < 0.05) relationship between a parameter and thermal perception, either positive (red circle) or negative (blue circle). Some parameters are not displayed, as they are too distant from the Van Dobben circles–indicating no significant relationship with thermal perception.

cause of the thermal perception differences. As the HI was the index with the strongest correlation with thermal perception, we present the HI values: the average HI in all three parks during the measured period was significantly (F = 47.4, p < $2 \times 10^{-16}$) affected by the daytime (23.6°C (± 4.5), 28.8°C (± 4.9), and 31.5°C (± 6.3), in the morning, afternoon, and evening, respectively) as well as the thermal perception. To explain better the reasons behind why HI correlates most closely with thermal perception, we performed a Pearson correlation analysis between the wind speed and visitors´ heat perception. The correlation was not significant at p < 0.05 (Pearson's $r$ = -0.14938).

## Discussion

This study evaluated the suitability of the use of four frequently used thermal comfort indices–WGBT, HI, PET, and UTCI–for the future planning of urban parks in Central Europe. Specifically, we analysed associations between the thermal indices and thermal perception of park visitors in three parks in Prague, and investigated whether or not thermal perception varies with regards to the parameters such as the visitors' sex, age, and activity, and between week-time and daytime. We conclude that all the indices reflect thermal perception well, while HI, showed the strongest correlation, followed by UTCI, PET and WBGT. We found that thermal perception was not dependent on the sex of park visitors and week-time of their visit; however, it was associated with the daytime of the visit, and slightly associated with the age and activity of the visitors.

### Association between the thermal indices and thermal perception of park visitors

Our first hypothesis that (i) all the indices would be associated with subjective thermal comfort, while the more complex indices would be more closely associated with thermal

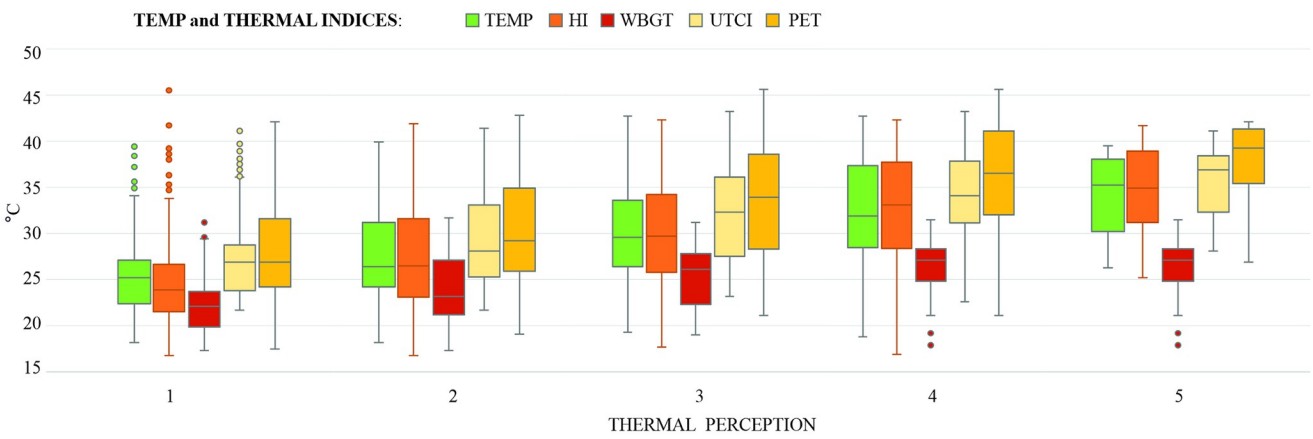

**Fig 4. Distribution of TEMP and thermal indices in the five thermal perception categories.**

perception, was partly supported by our data. All the indices reflect thermal perception well, while the simplest index, HI, was associated with thermal perception the most. It is the most surprising result as many studies confirmed the strong relationship between more comprehensive investigated indices, PET and UTCI and thermal perception (called thermal sensation vote in [29, 60, 64]).

This may to be due to the relative simplicity of HI–being based solely on air temperature (TEMP) and relative humidity, not taking wind speed and short-wave radiation into account, and not being based on the MEMI model. There is indeed a close relationship between TEMP at each level of the perception scale, which can be seen at Fig 4. Regarding the wind speed, it was not significantly correlated with thermal perception in our study, which was probably due to a small variability of wind speed during our survey (S1 Fig). In fact, some other studies [36, 39] found the wind speed influences thermal perception, especially in summer and winter [36]. In the Czech Republic, the wind speed in summer is typically lower than in autumn and winter (www.chmi.cz). This fact limits the applicability of HI to the conditions of Central Europe in the summer. Our results show that UTCI, also very closely related with thermal perception, can be calculated using the measured wind speed (instead of the usual one measured at 10 metres [19, 25]) without a significant effect on results. Therefore, it is not necessary to recalculate the wind speed to a height of 10 meters, as some studies do [65], not heaving the data recorded by meteorological stations. In more windy conditions, we could expect closer relation between UTCI and thermal perception [26], as the height of the measure device better reflects the nature of the wind at pedestrian level, unconstrained by terrain obstacles.

The short-wave radiation fluxes reaching the human body is more pronounced in summer [21, 66]. The effect of the radiation can differ both spatially and temporarily [60]. The short-wave radiation is represented by mean radiant temperature, present in other explored indices. The radiation in the in RayMan software, used for UTCI and PET calculation, is standardised [21], which tends to underestimate its effect [60]. The uncertainties regarding the mean radiant temperature, therefore, lead to WBGT, the least associated index with thermal perception. As such, the MEMI model set up in RayMan software [21], based on "universal" man's characteristics developed in Germany, proved to be fairly accurate for use in Central Europe. Nevertheless, minor imprecisions regarding physiological and psychological differences between people [36, 41, 51] are inevitable even if only local residents surveyed.

## Association between the thermal perception of park visitors and assessed parametres

Our second hypothesis, that (ii) thermal perception would be affected by sex, age, or the week-time or the daytime, was supported. Park visitors did not feel more comfortable on the week-ends than on weekdays. On the contrary, the visitors felt more comfortable in the mornings and less comfortable in the evenings. This can be largely explained meteorologically–the thermal indices increased towards the evenings (Fig 5a). Nevertheless, people tolerated higher temperatures in the evenings slightly better than in the mornings, which will probably be influenced by less strong solar radiation in the evenings than during the day. The radiation acts directly on the temperature receptors on the skin to produce a stronger sense of heat [51]. It may also be influenced by respondents' acclimatisation during the daytime [67], which underscores the fact that in the evenings there were mainly young people (young/middle-aged/elderly: 135/59/16) in the park, who adapt better to uncomfortable thermal conditions than elderly [42].

Consistent with many previous studies [19, 46, 52, 68], no significant difference in thermal perception range was found between men and women. In different environments and cultural

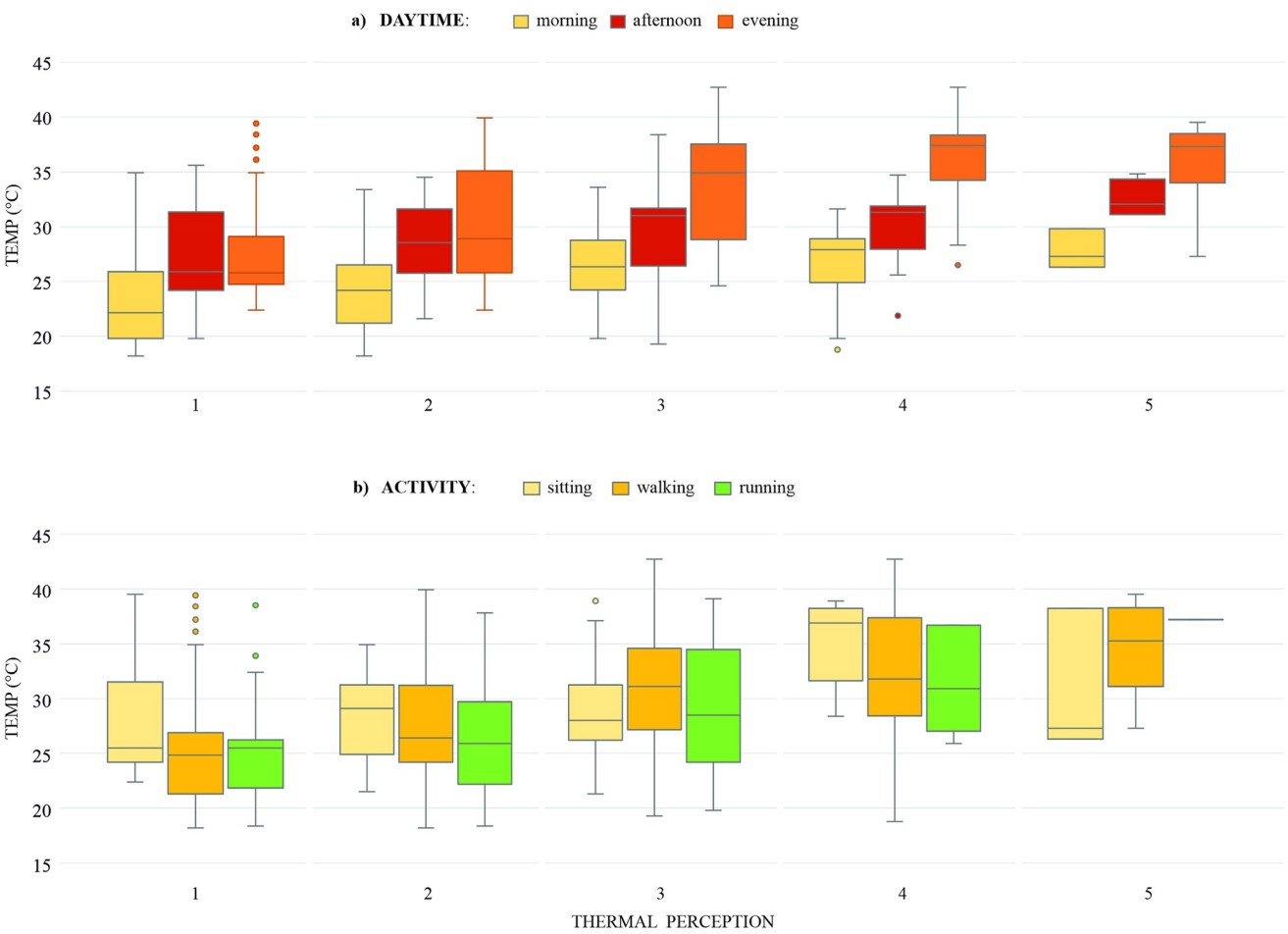

**Fig 5. Distribution of TEMP in the thermal perception categories (a) at different daytime, and (b) according to activities.**

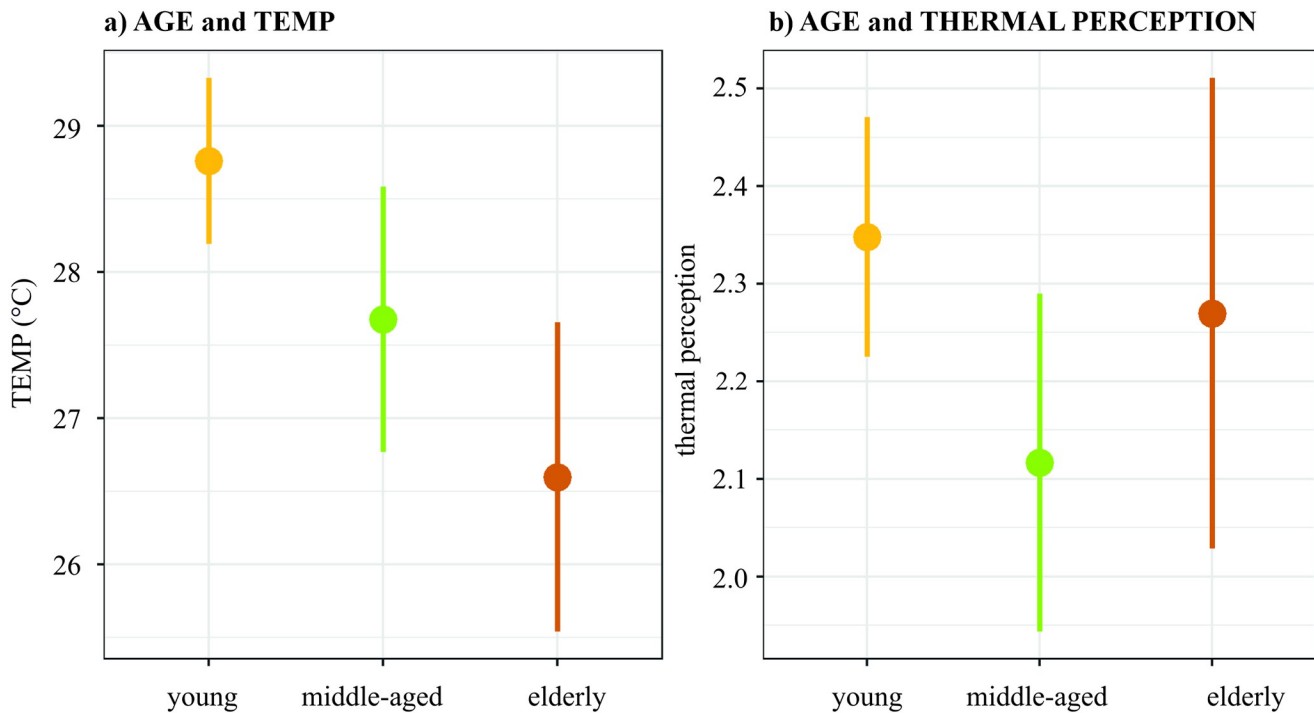

**Fig 6. Mean values of HI (a) and thermal perception (b) according to age of respondent.**

conditions, higher temperatures can sometimes be preferred by men [42] or women [49]. Such differences, however, can be compensated by different traditions of clothing [48].

In line with other studies [42], our results show that people performed more static activities at higher temperatures (Fig 5b), and this pattern was stronger with increasing age (young/middle-aged/elderly: 27.89˚C (±5.24)/ 25.39˚C (±5.09)/23.08˚C (±3.32)). They are not very surprising results; somewhat more surprising is that middle-aged visitors felt significantly more thermally comfortable than other age groups of visitors, while the low-active respondents (sitting and walking activities) felt significantly worse than other groups of visitors. A possible explanation can be in a combination of the effect of age and activity, which was found by some previous studies [46, 51]. Xiong and He [46] found the activity to have strong positive impact on thermal comfort regardless of weather conditions, and the impact is the strongest at middle-aged group (corresponding to 25–50 years in [46]). The positive effect of the activity showed our study as well, as the thermal perception of respondents performing high-active activities correlates with the air temperature less than the thermal perception of respondents performing static activities. In our study, the middle-aged group (87%) performed less low-active activities than elderly (94%) and more than young (81%). Stronger positive effect of activities on middle-aged group can, therefore, be a possible explanation in our study as well.

Elderly probably perform less activities and feel worse in connection with the metabolic rate decrease [48]. Temporal reasons are behind the fact that young people felt worse at high temperatures (Fig 6). When young, middle-aged, and elderly people visited the parks, the average temperature was 28.8˚C (±5.4), 27.7˚C (±4.5), and 26.6˚C (±5.0), respectively. The elderly, who generally struggle to cope with higher temperatures [42, 69], were performing static activities or were not present at all during periods of high temperatures. Despite elderly visited parks in average at lowest temperatures (Fig 6a), their thermal perception was worse than for

middle-aged (Fig 6b). We can therefore deduce that middle-aged group can tolerate the heat better than the elderly people. Finally, we can conclude that age and activity have a partial joint effect on thermal perception, but it is only little. Consistently, little age impact of age on thermal perception was found in the USA and China [48], as well as Poland [49]. In line with our research, the age and activity impact can be explained by a combination of reasons, such as the use of different methodology [29], the respondents' place of origin [49], or the location where the survey takes place [47]. This implies that, in practice, all age categories should be interviewed during the thermal perception survey, and the type of activity should be taken into consideration.

### Study limitations

The study has some limitations. The site survey was conducted in the summer season, covering a wide range of air temperatures, but with only moderate and stable wind speed, in similar landscape patterns in Central European climate zone. All these conditions affect thermal perception [11, 23, 46, 60]. For instance in a China´s cold region, meteorological parameters affect thermal perception much more in winter than summer [70]. We assume that in strong winds in winter people would feel colder and less comfortable [46]. Regarding landscape pattern, it is well known that greenery increases positive psychological effect on visitors´ perception, however, it might be lower in summer than in winter [67]. Therefore, our results are limited to the above-mentioned conditions. Hence, further research should also verify the results for a wider range of meteorological and geographical conditions throughout the year. It must also be noted that more parameters, such as the clothes of the respondents [39], should be considered when comparing different seasons.

In addition, slight discrepancies could be caused by the use of three different measurement devices in three parks, as well as five different interviewers. Nevertheless, the majority of measurements were performed by only one measuring device and one interviewer. Moreover, all interviewers were given detailed instructions before the actual interviews.

### Implications on urban planning and design

During the climate change, the outdoor thermal comfort must be considered to make public spaces more liveable, healthy, and comfortable. Weather changes are uncontrollable, while outdoor environments can be adjusted through principles of climate-sensitive planning and design. Our findings generated some implications on such principles for urban parks in the Central European climate during the summertime.

The results of our study suggest that all the indices can be used to describe links between thermal conditions and thermal perception, and that they are sufficient tools to describe thermal conditions in summer in the Central European climate. HI, the simple index, is particularly useful for those who are unfamiliar with complex thermal indices calculations, such as architects and urban planners, to create and evaluate climate-sensitive urban design. For instance, planners can measure the HI before and after a park intervention to evaluate it and use the results for setting conditions for the design of another park, e.g. in a design competition. No matter if the measurement is performed during the weekends or weekdays, but it should be carried out in the mornings and in the evenings to get revealing results.

Design strategies cooling down the air temperature can considerably increase thermal comfort in the parks in the summer. Opportunities for both low and high-active summer activities should be created in the parks regardless of the differences between women and men. Walking paths should not be exposed to the sun for their entire length but should be cooled, preferably by dense greenery and their shadow, during the whole day. Places for young people should be

cooled down especially in the evenings when young people use the parks the most. The number of benches should be increased, especially at places where the shade falls in the morning, as that is the time when the elderly often use parks for sitting. Design for the elderly should be highly appreciated by urban designers as they are the most sensitive to thermal comfort.

## Supporting information

**S1 Fig. Box plots of TEMP, humidity, wind speed.**
(TIF)

**S1 Table. Data.**
(XLSX)

## Acknowledgments

Thanks to Jaroslav Koudelka, Vojtěch Richter, Adéla Švecová, and Pavla Žebráková for performing the site survey.

## Author Contributions

**Conceptualization:** Vlaďka Kirschner, Aleš Urban.

**Data curation:** Lucie Chlapcová.

**Formal analysis:** Aleš Urban, Veronika Řezáčová.

**Investigation:** Vlaďka Kirschner.

**Supervision:** Veronika Řezáčová.

**Writing – original draft:** Vlaďka Kirschner.

**Writing – review & editing:** Aleš Urban, Veronika Řezáčová.

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
