## [Decision Letter · Decision Letter 0]

6 Dec 2023

PONE-D-23-36648

Thermal comfort perception among park users in Prague, Central Europe on hot summer days – a comparison of thermal indices

PLOS ONE

Dear Dr. Kirschner,

Thank you for submitting your manuscript to PLOS ONE. After careful consideration, we feel that it has merit but does not fully meet PLOS ONE’s publication criteria as it currently stands. Therefore, we invite you to submit a revised version of the manuscript that addresses the points raised during the review process.

We look forward to receiving your revised manuscript.

Kind regards,

Baojie He, Ph.D

Academic Editor

PLOS ONE

Journal Requirements:

   "VŘ was supported by the Ministry of Agriculture of the Czech Republic, institutional support MZE-RO0423. AU was supported by the Czech Academy of Sciences programme “Strategie AV21 – Dynamická planeta Země”. "

6. We note that Figure 1 in your submission contain map/satellite images which may be copyrighted. All PLOS content is published under the Creative Commons Attribution License (CC BY 4.0), which means that the manuscript, images, and Supporting Information files will be freely available online, and any third party is permitted to access, download, copy, distribute, and use these materials in any way, even commercially, with proper attribution. For these reasons, we cannot publish previously copyrighted maps or satellite images created using proprietary data, such as Google software (Google Maps, Street View, and Earth). For more information, see our copyright guidelines: http://journals.plos.org/plosone/s/licenses-and-copyright.

Reviewers' comments:

Reviewer's Responses to Questions

**Comments to the Author**

1. Is the manuscript technically sound, and do the data support the conclusions?

Reviewer #1: Partly

Reviewer #2: Yes

2. Has the statistical analysis been performed appropriately and rigorously? 

Reviewer #1: Yes

Reviewer #2: Yes

3. Have the authors made all data underlying the findings in their manuscript fully available?

Reviewer #1: Yes

Reviewer #2: Yes

4. Is the manuscript presented in an intelligible fashion and written in standard English?

Reviewer #1: Yes

Reviewer #2: Yes

5. Review Comments to the Author

Reviewer #1: The present manuscript rigorously investigates the correlation between the thermal comfort indices (WGBT, HI, PET and UTCI) and individuals' perception of thermal environments. It also explores the associations of gender, age, and other factors with thermal perception. While the manuscript represents a considerable investment of effort, in light of the prevailing extensive research focusing on the applicability of diverse indices, it is imperative that this study accentuates its innovative contributions to the field. In addition, several aspects/ questions should be addressed before considering to be published:

The manuscript would benefit from a more detailed exposition of the data analysis methodologies within the abstract. Furthermore, it is pertinent that the abstract's conclusions are specifically tailored to the summer season, rather than all seasons.

There is an omission of page numbers on the even-numbered pages.

Figure 1, it is imperative to enhance the presentation by incorporating the Sky View Factor (SVF) data for the three designated measurement points. Alternatively, the inclusion of photographs that vividly illustrate the environmental context of these points would be beneficial. Moreover, the figure, in its current form, falls short in terms of visual engagement and aesthetic quality. A thoughtful redesign is recommended, one that could benefit from adopting the advanced graphical methodologies as demonstrated in the following study: Huang, B.Z., Hong, B., Tian, Y., Yuan, T.T., Su, M.F., 2021. Outdoor thermal benchmarks and thermal safety for children: A study in China's cold region. Sci Total Environ 787. https://doi.org/10.1016/j.scitotenv.2021.147603.

The simultaneity of the experiments conducted in the three parks warrants clarification within the manuscript. In cases of non-simultaneous testing, the inclusion of varied meteorological parameter charts is necessary.

Figure 2 exhibits non-compliance with standard graphical representation, especially regarding the labeling of axes, necessitating a revision.

The content within Lines 157 – 164 would be more appropriately positioned subsequent to the computation of indices or within the results section.

Lines 176 – 177, the addition of bibliographic references is required.

The classification of walking as a static activity, as mentioned in Lines 190 – 191, appears to be a misclassification.

The manuscript's tables should adhere to the three-line table format.

Clarification is needed on the specific height at which wind speed measurements were taken for the UTCI calculations.

An explanation is warranted as to why, despite having collected data on activity levels, gender, and age in the questionnaire, standard values were employed in the RayMan software for the computation of PET or UTCI. The potential impact of this choice on the study's conclusions should also be addressed.

In the application of Pearson correlation analysis, it is crucial to confirm whether a test for normal distribution was conducted.

Regarding the CCA analysis, considerations such as the nature of the data, sample size, and the presence of multicollinearity need to be addressed.

The CCA's findings raise questions, particularly the rationale behind correlating separate age categories (middle-aged, young) with thermal perception and the implications of a negative correlation between middle age and thermal perception.

All figures in the manuscript require enhancements to improve their visual quality.

Dividing the discussion section into well-defined subsections would enhance the clarity and structure of the manuscript.

In light of the prevalent focus on the applicability of thermal comfort indices, the manuscript would be enriched by incorporating strategies pertaining to local planning, design, or thermal adaptation of the users.

The following papers may be of interest:

Xiong, K., He, B.J., 2022. Wintertime outdoor thermal sensations and comfort in cold-humid environments of Chongqing China. Sustainable Cities and Society 87. https://doi.org/10.1016/j.scs.2022.104203.

He, B.-J., Zhao, D., Dong, X., Xiong, K., Feng, C., Qi, Q., Darko, A., Sharifi, A., Pathak, M., 2022. Perception, physiological and psychological impacts, adaptive awareness and knowledge, and climate justice under urban heat: A study in extremely hot-humid Chongqing, China. Sustainable Cities and Society 79, 103685. https://doi.org/10.1016/j.scs.2022.103685.

Reviewer #2: General comments: I appreciate the author's efforts on the critical issue of selecting appropriate outdoor heat indicators for architects and planners in Central Europe. This article is a comprehensive and typical study of thermal comfort in Central Europe, as it comprehensively evaluates the applicability of 4 commonly used thermal indicators and considers the influence of multiple factors (gender/age/visit time/activity level, etc.) on the applicability of the indicators. However, there are still some errors in this article that require some modifications.

1. There are some errors in the number of respondents in Table1, e.g., in the Royal Preserve column, the total number of respondents is 187 when differentiated by week and weekend or men and women. But when differentiated by morning, afternoon and evening, the total number is 176. The same error occurs in the other two parks' respondent counts, so be sure to recheck the numbers and correct them.

2. In Fig.1, satellite images or aerial photos of the three parks can be added to help readers understand more detailed information about the sites (area, green space coverage, and surroundings, etc.).

3. In line 288, consider including a correlation analysis between wind speed and visitors' heat perception polls to determine if wind speed has an effect on visitors' heat perception? If the correlation is not strong, it suggests that the failure to account for changes in wind speed is the reason why the HI index is more competent for evaluating thermal comfort in Central European parks. A method for predicting the effect of a single meteorological factor on thermal perception can be found in this article: Factors influencing resident and tourist outdoor thermal comfort: A comparative study in China's cold region.

References:

Yu Tian, Bo Hong, Zhenqi Zhang, Shuang Wu, Tingting Yuan, Factors influencing resident and tourist outdoor thermal comfort: A comparative study in China's cold region, Science of The Total Environment, Volume 808, 2022, 152079, ISSN 0048-9697, https://doi.org/10.1016/j.scitotenv.2021.152079.

4. In line 322, people's tolerance for heat is higher at night than during the day, mainly because nighttime does not have the strong solar radiation of daytime, which acts directly on the temperature receptors on the skin to produce a stronger sense of heat. Of course, there can also be some influence of psychological factors such as thermal history that lead tourists to perceive hotter environments as more comfortable at night. You can refer to this article: Perception, physiological and psychological impacts, adaptive awareness and knowledge, and climate justice under urban heat: A study in extremely hot-humid Chongqing, China.

References:

Bao-Jie He, Dongxue Zhao, Xin Dong, Ke Xiong, Chi Feng, Qianlong Qi, Amos Darko, Ayyoob Sharifi, Minal Pathak, Perception, physiological and psychological impacts, adaptive awareness and knowledge, and climate justice under urban heat: A study in extremely hot-humid Chongqing, China, Sustainable Cities and Society, Volume 79, 2022, 103685, ISSN 2210-6707, https://doi.org/10.1016/j.scs.2022.103685.

5. Line 349, a) and b) should be used to illustrate the pictures in Fig. 6, not left or right.

6. Line 349, according to the results of Fig.6, it cannot be directly concluded that the middle-aged group can tolerate the heat better than the young people, but it can only be directly stated that the middle-aged group can tolerate the heat better than the old people. If there is any data analysis that can support the conclusion that the middle-aged group is the group that can tolerate heat the best, please add it.

6. PLOS authors have the option to publish the peer review history of their article (what does this mean?). If published, this will include your full peer review and any attached files.

Reviewer #1: No

Reviewer #2: No

---

## [Author Response · Author response to Decision Letter 0]

27 Jan 2024

Dear Reviewers

Thank you very much for your revisions and for your kind and helpful comments. 

Following your comments, we have integrated the following changes into the manuscript:

Editor:

Journal Requirements:

1. Please ensure that your manuscript meets PLOS ONE's style requirements, including those for file naming. The PLOS ONE style templates can be found at (address). 

Answer:

I have updated the format according to the recommended templates. 

Answer:

I have added the information in lines 176-178. 

 "VŘ was supported by the Ministry of Agriculture of the Czech Republic, institutional support MZE-RO0423. AU was supported by the Czech Academy of Sciences programme "Strategie AV21 – Dynamická planeta Země". "

Answer:

First, I have deleted information about the grant from the Acknowledgments as it was said so in PLOS ONE style templates. Second, I added the recommended sentence to the text (lines 527-529). I added the sentence to the cover letter as well. 

Answer:

There is no DOI of the data; the data has not been published. However, we offer them as open data as part of this article. We added an Excel file with all the data as Supporting information (S2). We have mentioned it in the cover letter as well. 

5. Please include your full ethics statement in the 'Methods' section of your manuscript file. In your statement, please include the full name of the IRB or ethics committee who approved or waived your study, as well as whether or not you obtained informed written or verbal consent. If consent was waived for your study, please include this information in your statement as well. 

Answer:

We have added the full name of the Committee to the Methods (lines 176-177). The information about park visitors' consent is in the subsequent text (lines: 179-181). 

6. We note that Figure 1 in your submission contain map/satellite images which may be copyrighted. All PLOS content is published under the Creative Commons Attribution License (CC BY 4.0), which means that the manuscript, images, and Supporting Information files will be freely available online, and any third party is permitted to access, download, copy, distribute, and use these materials in any way, even commercially, with proper attribution. For these reasons, we cannot publish previously copyrighted maps or satellite images created using proprietary data, such as Google software (Google Maps, Street View, and Earth). For more information, see our copyright guidelines: (guidelines) 

Answer:

Thank you very much for the detail guidelines. We used Czech open data. We added information about them under the Fig 1 (lines 154-155). 

7. Please include captions for your Supporting Information files at the end of your manuscript, and update any in-text citations to match accordingly. Please see our Supporting Information guidelines for more information (web page). 

Answer:

I renamed the supporting files (S1 and S2) and added the information to the text (lines 747-749) according to the guidelines.  

Reviewer #1: 

Comment: The manuscript would benefit from a more detailed exposition of the data analysis methodologies within the abstract. Furthermore, it is pertinent that the abstract's conclusions are specifically tailored to the summer season, rather than all seasons.

Answer:

The data analysis methodology and a reference to summer season has been added to the abstract. 

Comment: There is an omission of page numbers on the even-numbered pages.

Answer:

The page numbering has been corrected. 

Comment: Figure 1, it is imperative to enhance the presentation by incorporating the Sky View Factor (SVF) data for the three designated measurement points. Alternatively, the inclusion of photographs that vividly illustrate the environmental context of these points would be beneficial. Moreover, the figure, in its current form, falls short in terms of visual engagement and aesthetic quality. A thoughtful redesign is recommended, one that could benefit from adopting the advanced graphical methodologies as demonstrated in the following study: Huang, B.Z., Hong, B., Tian, Y., Yuan, T.T., Su, M.F., 2021. Outdoor thermal benchmarks and thermal safety for children: A study in China's cold region. Sci Total Environ 787. https://doi.org/10.1016/j.scitotenv.2021.147603. 

Answer:

Thank you for a demonstrative example of the figure. We got inspired and redesigned the Figure 1. We incorporated the orthophoto map with surroundings and pictures with people (with an attempt to show unrecognizable characters) to show the context better. 

Comment: The simultaneity of the experiments conducted in the three parks warrants clarification within the manuscript. In cases of non-simultaneous testing, the inclusion of varied meteorological parameter charts is necessary.

Answer:

Yes, the data in the three parks were collected in different days and times (Royal Preserve: 12./14.6.2022, 21./22./24.7.2022, 25./27.6.2023, 11.15./7.2023; Central Park: 11./12./14.6.2022, 21./22./24./25.7.2022, 25./28.6.2023, 2.7.2023, Hvězda Preserve: 21./22./23./24./25./27.6.2023, 2./9./14.7.2023; the data are part of an excel table as the open source). According to your recommendation, the meteorological parameters, such as air temperature, humidity, and wind speed has been displayed in charts (boxplots) in the Supplement 1. The reference to the Supplement is in Methods / Discussion (lines 218, 357).

Comment: Figure 2 exhibits non-compliance with standard graphical representation, especially regarding the labeling of axes, necessitating a revision. 

Answer:

The graphical representation of the graph in Figure 2, such as the representation of all the graphs in the text, has been uniformed and upgraded. 

Comment: The content within Lines 157 – 164 would be more appropriately positioned subsequent to the computation of indices or within the results section.

Answer:

First, we have removed two sentences about the meteorological data measurement at the end of the same chapter (Field survey). Second, we have removed the text in lines 157 – 164 to the second paragraph of subsequent chapter (Indices calculation), before UTCI and PET data. It fits much better into the text now. Thank you for noticing. 

Comment: Lines 176 – 177, the addition of bibliographic references is required. 

Answer:

Thank you for the notification. The reference (Daniel Kahneman et al., 2021: Noise: a flaw in human judgment, no 58) has been added to the text. 

Comment: The classification of walking as a static activity, as mentioned in Lines 190 – 191, appears to be a misclassification.

Answer:

We agree that the term was inappropriate. We changed the term "static" activity to "low-intensive physical activity" (in short: low-active), the term "dynamic" activity to "high-intensive physical activity" (in short: high-active). It seems more appropriate now. Thank you for noticing. 

Comment: The manuscript's tables should adhere to the three-line table format.

Answer:

The table format has been adjusted to the three-line table format. 

Comment: Clarification is needed on the specific height at which wind speed measurements were taken for the UTCI calculations.

Answer:

The height of wind speed for the calculation is indeed important, and we did not mention it in the methods; you are right. The UTCI calculation height was the same as the measurement height – 1.5 meters. Heaving measured the wind speed; we did not need more genderized wind speed from meteorological stations (measured at 10 meters in height). Before starting the investigation, we had considered both options. One of the reasons we decided to go with the first option was the moderate wind conditions, which indicated that the wind would not have much effect on the results (and, indeed, the correlation between the wind speed and perception was not significant). We have clarified it in the Methods (line 255) and discussed implications in the Discussion (lines 355-366). 

Comment: An explanation is warranted as to why, despite having collected data on activity levels, gender, and age in the questionnaire, standard values were employed in the RayMan software for the computation of PET or UTCI. The potential impact of this choice on the study's conclusions should also be addressed.

Answer:

It is a good question. Thank you for it. We had discussed it before we started the research, and we decided to test the indices in RayMan software standard settings. We had two reasons for this decision. First, we obtained standard values based on complex physiological models with specific reference conditions including a reference person. Using the reference person characteristics, we can better distinguish how well the individual thermal perception of each respondent corresponds with the expected perception. Second, the software does not allow to see the calculation algorithm. The reference to the MEMI model as standard settings has been mentioned in the Methods (lines 230-234) and the Discussion (lines 389-393). 

The question may arise whether the reference settings of the indices are relevant for the real live situations. But answering this question is beyond the aim of this study.

Comment: In the application of Pearson correlation analysis, it is crucial to confirm whether a test for normal distribution was conducted.

Answer:

The test for normal data distribution had been performed before the correlation analysis. This information was added to the Methods / Data analyses (line 253).

Comment: Regarding the CCA analysis, considerations such as the nature of the data, sample size, and the presence of multicollinearity need to be addressed.

Answer:

Thank you for the notification. The pack of analyzed data contained 600 respondents (samples) and all the assessed parameters (species; see above), which were as a qualitative (categorical) data coded as dummy (0/1) variables, and 1 explanatory variable (thermal perception) The mentioned points have been addressed in the Methods / Data analyses (lines 259-270). 

Comment: The CCA's findings raise questions, particularly the rationale behind correlating separate age categories (middle-aged, young) with thermal perception and the implications of a negative correlation between middle age and thermal perception. 

Answer:

Creating the age categories, we based on sociological division by Erikson (1950): Childhood and Society – added to the text, no 59), who divides the life cycle according to biological processes into age up to 30, 30-60, and more than 60. When we estimated the age of respondents in the survey, we rounded to 5 years. Age categories were then correlated separately due to its categorical nature which require coding the data as dummy (0/1) variables. Indeed, we agree that the negative correlation between middle age and thermal perception is interesting and worthy discussion. We have used the recommended reference and discussed it more deeply in the Discussion / Association between the thermal perception of park visitors and assessed parametres. 

Comment: All figures in the manuscript require enhancements to improve their visual quality.

Answer: 

We have redesigned all charts in an attempt to improve their quality. 

Comment: Dividing the discussion section into well-defined subsections would enhance the clarity and structure of the manuscript.

In light of the prevalent focus on the applicability of thermal comfort indices, the manuscript would be enriched by incorporating strategies pertaining to local planning, design, or thermal adaptation of the users.

Answer: 

We have divided the Discussion into four sub-sections while the last one concentrates on the application of the results. The strategies pertaining to local planning and design were incorporated into this subsection: Implications on urban planning and design. In doing so, we deleted a paragraph with unnecessary information, which did not fit into the structured text (lines 369-383). 

Comment: The following papers may be of interest:

Xiong, K., He, B.J., 2022. Wintertime outdoor thermal sensations and comfort in cold-humid environments of Chongqing China. Sustainable Cities and Society 87. https://doi.org/10.1016/j.scs.2022.104203.

He, B.-J., Zhao, D., Dong, X., Xiong, K., Feng, C., Qi, Q., Darko, A., Sharifi, A., Pathak, M., 2022. Perception, physiological and psychological impacts, adaptive awareness and knowledge, and climate justice under urban heat: A study in extremely hot-humid Chongqing, China. Sustainable Cities and Society 79, 103685. https://doi.org/10.1016/j.scs.2022.103685.

Answer: 

Thank you for these useful references. We have used them to discuss the results deeply. 

Reviewer #2: 

1. Comment: There are some errors in the number of respondents in Table1, e.g., in the Royal Preserve column, the total number of respondents is 187 when differentiated by week and weekend or men and women. But when differentiated by morning, afternoon and evening, the total number is 176. The same error occurs in the other two parks' respondent counts, so be sure to recheck the numbers and correct them. 

Answer:

All errors in Table 1 have been corrected and controlled twice. 

1. Comment: In Fig.1, satellite images or aerial photos of the three parks can be added to help readers understand more detailed information about the sites (area, green space coverage, and surroundings, etc.).

Answer:

Three orthophoto maps and three photos have been added to Figure 1 to show more information about the sites and their surroundings.

3. Comment: In line 288, consider including a correlation analysis between wind speed and visitors' heat perception polls to determine if wind speed has an effect on visitors' heat perception? If the correlation is not strong, it suggests that the failure to account for changes in wind speed is the reason why the HI index is more competent for evaluating thermal comfort in Central European parks. A method for predicting the effect of a single meteorological factor on thermal perception can be found in this article: Factors influencing resident and tourist outdoor thermal comfort: A comparative study in China's cold region.

Answer:

We have done the correlation between wind speed heat perception and added the results to the text (lines 316-318). The correlation was not significant. At the same time, the survey was performed during very moderate and stable wind conditions. We reflected it in the Discussion / Association between the thermal indices and thermal perception of park visitors. Thank you for an interesting and inspiring reference; we refer to it in the article (no 36). 

4. Comment: In line 322, people's tolerance for heat is higher at night than during the day, mainly because nighttime does not have the strong solar radiation of daytime, which acts directly on the temperature receptors on the skin to produce a stronger sense of heat. Of course, there can also be some influence of psychological factors such as thermal history that lead tourists to perceive hotter environments as more comfortable at night. You can refer to this article: Perception, physiological and psychological impacts, adaptive awareness and knowledge, and climate justice under urban heat: A study in extremely hot-humid Chongqing, China.

Answer:

Thank you for the very relevant point of the strong solar radiation. I took the liberty of using your clearly explanatory words, using also the recommended article. We omitted thermal history reasons as foreigners and people arriving from different thermal environment were not included in our study. 

5. Comment: Line 349, a) and b) should be used to illustrate the pictures in Fig. 6, not left or right. 

Answer:

a) and b) was added to illustrate the picture in Figure 6. 

6. Comment: Line 349, according to the results of Fig.6, it cannot be directly concluded that the middle-aged group can tolerate the heat better than the young people, but it can only be directly stated that the middle-aged group can tolerate the heat better than the old people. If there is any data analysis that can support the conclusion that the middle-aged group is the group that can tolerate heat the best, please add it. 

Answer:

We agree that Fig. 6 does not imply this conclusion directly. Figure 6 shows that only the middle-aged group has a lower values expressing thermal perception. However, it can only be deduced that since this group reduced (negatively correlated) thermal perception values at the same range of temperature conditions as those set for young and elderly people (who had no effect on thermal perception, Fig. 3) the middle-aged group was more heat tolerant. Therefore, we have included a reference to Figure 6 b, which shows this reduction directly. We extended discussion about this topic in the Discussion / Association between the thermal perception of park visitors and assessed parametres. 

Thank you again for your comments and recommendations. They helped improve quality of the article. 

We believe our changes will be to your satisfaction. 

With kind regards, 

Corresponding author

---

## [Decision Letter · Decision Letter 1]

9 Feb 2024

Thermal comfort perception among park users in Prague, Central Europe on hot summer days – a comparison of thermal indices

PONE-D-23-36648R1

Dear Dr. Kirschner,

We’re pleased to inform you that your manuscript has been judged scientifically suitable for publication and will be formally accepted for publication once it meets all outstanding technical requirements.

Kind regards,

Baojie He, Ph.D

Academic Editor

PLOS ONE

Additional Editor Comments (optional):

Reviewers' comments:

Reviewer's Responses to Questions

**Comments to the Author**

1. If the authors have adequately addressed your comments raised in a previous round of review and you feel that this manuscript is now acceptable for publication, you may indicate that here to bypass the “Comments to the Author” section, enter your conflict of interest statement in the “Confidential to Editor” section, and submit your "Accept" recommendation.

Reviewer #1: All comments have been addressed

2. Is the manuscript technically sound, and do the data support the conclusions?

Reviewer #1: Yes

3. Has the statistical analysis been performed appropriately and rigorously? 

Reviewer #1: Yes

4. Have the authors made all data underlying the findings in their manuscript fully available?

Reviewer #1: Yes

5. Is the manuscript presented in an intelligible fashion and written in standard English?

Reviewer #1: Yes

6. Review Comments to the Author

Reviewer #1: Thank you for addressing the concerns raised in my previous review. I have reviewed the revised manuscript and find it satisfactory. I have no further comments at this time. Thank you for your attention to my feedback.

7. PLOS authors have the option to publish the peer review history of their article (what does this mean?). If published, this will include your full peer review and any attached files.

Reviewer #1: No

---

## [Editor Report · Acceptance letter]

11 Mar 2024

PONE-D-23-36648R1 

PLOS ONE

Dear Dr. Kirschner, 

I'm pleased to inform you that your manuscript has been deemed suitable for publication in PLOS ONE. Congratulations! Your manuscript is now being handed over to our production team.

Kind regards, 

on behalf of

Dr. Baojie He 

Academic Editor

PLOS ONE